# A Large Language Model-based Bandwidth Prediction Algorithm

**Zheng Jiang**
2024310655

**Iat Long long**
2024270018

**Xin Chen**
2024310679

## 1   Background

In the streaming media industry, bandwidth prediction is essential for ensuring user experience and optimizing resource use. In low-latency live streaming, it estimates user network conditions in real-time, adjusts transmission strategies, and reduces stuttering and latency. For long videos, it helps adaptive algorithms select bitrates intelligently, adapting to network changes and balancing picture quality, smoothness, and buffering. In short videos, it determines video bitrate combinations for seamless and high-definition (HD) playback, enhancing user retention. Bandwidth prediction also affects the cost and efficiency of CDN distribution by optimizing transcoding bitrates and scheduling. Accurate bandwidth prediction drives decision-making algorithms, optimizes user experience and technical architecture, and is crucial for competitiveness in the streaming media industry.

User demands and network complexities make bandwidth prediction challenging, and traditional algorithms struggle with long-term bandwidth variations. Recently, Transformer and Large Language Models (LLMs) from the field of time-series prediction offer new solutions. Transformers excel with their strong feature extraction and long-term dependency modeling abilities, while LLMs adapt quickly to new tasks using pre-training on vast data. Applying these models to bandwidth prediction can significantly enhance accuracy, generalization, and real-time performance, better addressing the challenges of bandwidth prediction.

## 2   Problem Definition

Our project's significance resides in the development of a foundational bandwidth prediction algorithm tailored to accommodate diverse business scenarios, realizing bandwidth forecasting capabilities that span from millisecond to daily granularity through a unified modeling framework, thereby fulfilling the bandwidth prediction requirements of various applications, including low-latency live streaming, short/long video-on-demand, CDN, and others, ultimately leading to improved user experience.

Specifically, the research scope of this project encompasses two primary components:

### 2.1   Construct a benchmark that aligns with the real-world online bandwidth distribution

The construction of a suitable benchmark is a crucial prerequisite for training a bandwidth prediction algorithm. The collection and quantity of online data required to accurately represent online distributions are critical factors in creating an offline dataset that mirrors real-world online environments. Moreover, a unified framework that caters to diverse business scenarios demands a standardized input framework, which involves analyzing application-layer input features and fine-grained network-layer parameters to provide an exhaustive range of input information.

### 2.2   Bandwidth Prediction Algorithm for Long-Term and Short-Term Forecasting

To meet the diverse granularity requirements for bandwidth prediction across various business scenarios, a sophisticated approach is essential. Low-latency live streaming requires millisecond-

Submitted to Tsinghua University Course: Advanced Machine Learning (AML 2024). Do not distribute.

scale precision, while long/short video rate adaptive algorithms necessitate predictions on the order of seconds to minutes. Conversely, CDN operations benefit from forecasts with hourly to daily granularity. Therefore, a bandwidth prediction algorithm must effectively balance precise short-term dynamism awareness with long-term dependencies modeling.

Generally, **Bandwidth Prediction (BP)** can be formally defined as:

$$P(\mathbf{b}_{t+1:t+H}|\mathbf{b}_{1:t}) = \prod_{h=1}^{H} P(\mathbf{b}_{t+h}|\mathbf{b}_{1:t+h-1}) \tag{1}$$

where $\mathbf{b}_{1:t}$ represents the historical bandwidth usage data up to time $t$, $\mathbf{b}_{t+h}$ is the predicted bandwidth usage at future time $t + h$, and $H$ is the forecast horizon, indicating the extent of future predictions.

## 3 Proposed Method

### 3.1 Benchmark Construction via Tail Importance Sampling

Our approach involves constructing an offline dataset through tail importance sampling of online data, with a focus on the tail regions (characterized by low bandwidth) to ensure representativeness of the online data distribution. We consider traditional bandwidth representation states, such as download volume and time, and also seek to incorporate additional network-layer information from congestion control algorithms to enhance input informativeness. To accommodate multi-task scenarios, we create datasets from three distinct scenarios: video-on-demand (VOD), live streaming, and real-time communication (RTC). Specifically, our VOD dataset is chunk-based, while our live streaming and RTC datasets capture frame-level data, including size and transmission time, thereby enabling the construction of bandwidth datasets at varying granularities.

### 3.2 Bandwidth Prediction Based on Large Language Model

Our objective is to harness the capabilities of large language models to model the temporal dependencies of bandwidth, thereby achieving superior short-term and long-term predictive performance. Through an empirical evaluation of existing time series forecasting algorithms on bandwidth datasets, including TimeMixer, PatchTST, and TimeLLM [6, 5, 3], we investigate the task-specific characteristics of bandwidth prediction and the relative strengths and weaknesses of current approaches.

Presently, we propose to adopt TimeLLM [3] as the foundation model and perform domain-specific fine-tuning on the bandwidth dataset. By accounting for the temporal properties of bandwidth, we re-encode the input data and utilize self-attention mechanisms on the embedded variable tokens to capture the relationships between variables, thereby enhancing the predictive capabilities of the model.

## 4 Related Work

Network transmission time series prediction is essential for forecasting future network traffic and latency using historical data, aiming to optimize resource management. Recent advancements include linear model-based methods like TiDE, N-Hits, and Dlinear [1, 2, 8], which use various forms of linear regression. TiDE excels with simple datasets [2], N-Hits improves accuracy with cyclic data [1], and Dlinear adapts to rapid changes [8]. Transformer-based approaches, leveraging deep learning, include PatchTST, FEDformer, Pyraformer, Autoformer, and Informer [4, 5, 7, 9, 10]. These methods use intricate mechanisms like segmented data patches [5], combined attention mechanisms [10], and sparse attention for efficient long-term predictions and handling complex data patterns [7, 9]. Linear models are simple and computationally efficient, while Transformer-based methods excel with complex time series data. Future research may explore hybrid models that combine the strengths of both approaches for better accuracy and efficiency.

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
