# OpenReview forum: "【Proposal】A Large Language Model-based Bandwidth Prediction Algorithm"
_tsinghua.edu.cn/THU/2024/Fall/AML — THU 2024 Fall AML Submission_

### Official Review · ~Guilherme_Félix_Diogo1 · 2024-11-06
**Complete proposal, almost no flaws**

**Rating:** 9
**Confidence:** 4

**Review:**

This proposal introduces the large language models (LLMs) application for predicting bandwidth over different time periods in order to facilitate better resource management and improve the experience of users engaging in streaming media content. The approach is comprehensive and includes a benchmark for real-world bandwidth distribution as well as short and long-range predictions, and covers several industry scenarios from live bandwidth streaming to CDN based optimization. The proposed benchmark construction through tail importance sampling and application of TimeLLM, which has been fine tuned for bandwidth predictions, are highly practical and aim at addressing both the short term variations and the long term changes of the bandwidth data over the time. Could be a little more detailed on how it intends to assess the predictive performance across different contexts, particularly with respect to adjusting in real time.

---

### Official Review · ~Bowen_Su1 · 2024-11-08
**Good and Comprehensive Proposal**

**Rating:** 9
**Confidence:** 4

**Review:**

A comprehensive introduction was given to the benchmark construction involved in the task and the conceptual framework of the model. In addition, a detailed and comprehensive literature review has been conducted on this issue.Fully meets the requirements of the proposal, good work.

---

### Official Review · ~Zhen_Leng_Thai1 · 2024-11-08
**Comprehensive Proposal regarding Bandwidth Prediction based on LLM**

**Rating:** 9
**Confidence:** 4

**Review:**

This paper presents an LLM-based bandwidth prediction model for diverse streaming scenarios, supported by a robust benchmark dataset. Definition and methodology are well-defined, leveraging TimeLLM for real-time accuracy. However, more citations in the background section and reasoning for choosing TimeLLM could be beneficial.

---

### Official Review · ~Guanglei_He1 · 2024-11-10
**The overall proposal is relatively clear, establishing an benchmark dataset is likely to be extremely challenging.**

**Rating:** 8
**Confidence:** 4

**Review:**

Overall, The entire proposal to be exceptionally well-written—concise and clear. It is currently very lucid, and it appears that substantial effort has been invested in describing related work.

After a comprehensive review, I have a few suggestions:

1. **Benchmark Dataset Establishment**

   I understand that the most critical aspect of this work lies in establishing a benchmark dataset. Having this dataset as a foundation is essential to validate the effectiveness of subsequent methods.

   As I am not deeply familiar with this field, I am uncertain about the difficulty involved in collecting this data. The proposal mentions collecting data from three different scenarios, and there are several key considerations:

   - **Specific Application Data**: Is the data being collected from certain specific applications? The characteristics of network data streams from different applications are likely to be entirely different, which could result in fundamentally different models or methods. I believe that such application types should be explicitly specified in this proposal.
   - **Scope of Data Collection**: What is the extent of the data collection? For example, considering WeChat, its data is extremely widespread, distributed across national nodes, and much of it can be processed locally. Comprehensive data collection would be a highly challenging task. If the data is not comprehensive, could this negatively impact the overall effectiveness of the application?

2. **Choice of Models in Related Work**

   The Related Work section mentions that the industry predominantly uses linear model-based models due to their simplicity and efficiency. However, this proposal directly employs Large Language Models (LLMs). I personally feel that using LLMs in this context might be overly heavy. If the goal is merely to extract better features, I believe traditional models like CNNs and LSTMs could achieve an excellent balance between efficiency and effectiveness in this scenario. How has this been considered?

---

### Official Review · ~Keyu_Shen1 · 2024-11-10
**Clear and Well-organized Proposal**

**Rating:** 8
**Confidence:** 3

**Review:**

The background of proposed work is clearly stated in the proposal, and problem definition is effectively established. Main goals include benchmark constructing and TimeLLM-based framework developing, which has great potential of advancing cutting-edge AI tools in bandwidth prediction. However, the proposal would benefit from a more detailed discussion on computational resource management and response speed optimization, as the high computational demands of LLMs may pose challenges for real-time performance.

---

### Official Review · ~Yu_Zhang61 · 2024-11-11
**Review of "A Large Language Model-based Bandwidth Prediction Algorithm"**

**Rating:** 8
**Confidence:** 4

**Review:**

This thesis proposal introduces a promising approach for improving bandwidth prediction in the streaming media industry by leveraging large language models (LLMs) and Transformer architectures. The use of LLMs for bandwidth prediction is innovative, as these models have demonstrated success in extracting complex features and managing long-term dependencies, which could significantly enhance the accuracy and responsiveness of bandwidth predictions in diverse streaming contexts. The proposal is well-motivated, addressing the practical implications of improved bandwidth prediction, such as enhanced user experience through better bitrate selection and lower CDN costs due to optimized bitrate scheduling. However, while the potential benefits are clear, the proposal would benefit from a more detailed discussion of the specific challenges and complexities of adapting LLMs to bandwidth prediction over other established time-series models. A more concrete outline of the proposed architecture and evaluation metrics, as well as consideration of computational costs associated with deploying LLMs in real-time prediction contexts, would enhance the proposal's feasibility and clarity.
- I hope the author will elaborate on the detailed role and specific implementation of vit in this framework.

---

### Official Review · ~Chumeng_Jiang1 · 2024-11-12
**Well-structured, but some details still need to be supplemented.**

**Rating:** 8
**Confidence:** 3

**Review:**

This proposal puts forward the idea of leveraging large language models (LLMs) to design a foundational bandwidth prediction algorithm applicable to multiple scenarios, such as long-term and short-term live streaming. To achieve this goal, the authors expect to first construct a benchmark using tail importance sampling, and then propose "TimeLLM" to finetune the LLM on this benchmark dataset.

Strengths:
- **Well-structured and well-planned:** The proposal has a clear logical structure, with a defined problem, methodology, and planned steps for implementation.
- **The problem is significant and ambitious:** The aim is to develop a foundational bandwidth prediction algorithm for diverse scenarios with varying characteristics.

Weaknesses:
- **Insufficient details on how LLMs will be applied, with unclear necessity for LLMs:** It’s still unclear how the authors plan to utilize LLMs. Is it intended to transform all features into natural language and input them into the LLM? If that’s the case, how would this approach enable "harnessing the capabilities of large language models to model the temporal dependencies of bandwidth"?

---

### Official Review · ~Eddy_Yue1 · 2024-11-12
**Good idea!**

**Rating:** 9
**Confidence:** 4

**Review:**

Overall, the project presents a promising solution to bandwidth prediction challenges using advanced modelling techniques. Using tail importance sampling to enhance dataset accuracy focusing on low-bandwidth scenarios aligns well with the project's objectives. A contingency plan could strengthen the model's adaptability.

---

### Official Review · ~ChenJian1 · 2024-11-12
**Highly Innovative**

**Rating:** 9
**Confidence:** 4

**Review:**

The proposal presents a bandwidth prediction algorithm based on Large Language Models (LLMs) aimed at improving user experience and resource optimization in the streaming media industry. The core of the project is to develop a bandwidth prediction algorithm that can accommodate various business scenarios, achieving bandwidth forecasting capabilities from millisecond to daily granularity. The research scope includes constructing a benchmark test aligned with the real-world online bandwidth distribution and developing a bandwidth prediction algorithm for long-term and short-term forecasting.

# Strengths:
### ①Innovativeness:
The proposal to apply Large Language Models to bandwidth prediction is a novel attempt that has the potential to significantly improve the accuracy and generalization of predictions.
### ②Comprehensiveness:
The project considers the bandwidth prediction needs under different business scenarios, from millisecond to daily levels, showing the wide applicability of the algorithm.
### ③Data-driven:
The construction of the benchmark test through tail importance sampling enhances the model's representativeness of low-bandwidth areas, which is crucial for improving prediction accuracy.

# Weaknesses:
### ①Model Complexity:
Large Language Models may require substantial computational resources, which could limit the algorithm's practicality in resource-constrained environments.
### ②Real-time Performance:
The discussion on the model's real-time performance in the proposal is not detailed enough, especially in low-latency live streaming scenarios.

---

### Official Review · ~Kairong_Luo1 · 2024-11-12
**Novel idea**

**Rating:** 9
**Confidence:** 4

**Review:**

Strength:
1. The perspective is novel and have practical requirement;
2. The problem definition is clean, including how to measure the performance, the problem format;
3. There are clean steps to build benchmark and implement the algorithm;

Weakness:
1. The problem feature is unclear, like what is the difference between the bandwidth prediction from the other time series prediction, like stock market curve?
2. More details needed, like dataset collection. It seems non-trivial.

---

### Official Review · ~Justinas_Jučas3 · 2024-11-12
**Clear and Well-Structured Proposal with Clear Disadvantages**

**Rating:** 7
**Confidence:** 4

**Review:**

The proposal is written on a new algorithm of a well-established an relevant problem. It contains well-written, original and clear idea for solution, however, in my eyes, it is not clear why the proposed solution is not an overkill for the given problem. In addition, several key requirements are not satisfied.

## Advantages
1. A well-structured and easy-to-read proposal
2. The proposed algorithm is rather unique and clear
3. Most of the requirements are fulfilled

## Disadvantages
1. References to concrete datasets used are missing (this was a requirement!)
2. No specific details of how the performance is evaluated (what metrics can/will be used) are provided (this was a requirement!)
3. I do not see why a LLM is more beneficial than using for instance a simple NN. Why not use some regular time series estimation technique? Is LLM not an overkill for this task?
4. Lack of references. For example, many fact statements in the background section are not based on any sources, which is bad practice.

---

### Official Review · ~Kuanghao_Wang1 · 2024-11-12
**Interesting application direction**

**Rating:** 9
**Confidence:** 3

**Review:**

This PROPOSAL explains well the significance of the study that it is necessary and valuable to use machine learning for bandwidth prediction. Also, this PROPOSAL gives the specific problem and the methodology of prediction with a more extensive literature research. A slight shortcoming is that this paper does not give an explanation as to why TimeLLM is used as a method for prediction. It might be better if this aspect can be reinforced.